# Unified Methods for Exploiting Piecewise Linear Structure in Convex Optimization

**Tyler B. Johnson**
University of Washington, Seattle
tbjohns@washington.edu

**Carlos Guestrin**
University of Washington, Seattle
guestrin@cs.washington.edu

## Abstract

We develop methods for rapidly identifying important components of a convex optimization problem for the purpose of achieving fast convergence times. By considering a novel problem formulation—the minimization of a sum of piecewise functions—we describe a principled and general mechanism for exploiting piecewise linear structure in convex optimization. This result leads to a theoretically justified working set algorithm and a novel screening test, which generalize and improve upon many prior results on exploiting structure in convex optimization. In empirical comparisons, we study the scalability of our methods. We find that screening scales surprisingly poorly with the size of the problem, while our working set algorithm convincingly outperforms alternative approaches.

## 1 Introduction

Scalable optimization methods are critical for many machine learning applications. Due to tractable properties of convexity, many optimization tasks are formulated as convex problems, many of which exhibit useful structure at their solutions. For example, when training a support vector machine, the optimal model is uninfluenced by easy-to-classify training instances. For sparse regression problems, the optimal model makes predictions using a subset of features, ignoring its remaining inputs.

In these examples and others, the problem's "structure" can be exploited to perform optimization efficiently. Specifically, given the important components of a problem (for example the relevant training examples or features) we could instead optimize a simpler objective that results in the same solution. In practice, since the important components are unknown prior to optimization, we focus on methods that rapidly discover the relevant components as progress is made toward convergence.

One principled method for exploiting structure in optimization is *screening*, a technique that identifies components of a problem guaranteed to be irrelevant to the solution. First proposed by [1], screening rules have been derived for many objectives in recent years. These approaches are specialized to particular objectives, so screening tests do not readily translate between optimization tasks. Prior works have separately considered screening irrelevant features [1–8], training examples [9, 10], or constraints [11]. No screening test applies to all of these applications.

*Working set* algorithms are a second approach to exploiting structure in optimization. By minimizing a sequence of simplified objectives, working set algorithms quickly converge to the problem's global solution. Perhaps the most prominent working set algorithms for machine learning are those of the LIBLINEAR library [12]. As is common with working set approaches, there is little theoretical understanding of these algorithms. Recently a working set algorithm with some theoretical guarantees was proposed [11]. This work fundamentally relies on the objective being a constrained function, however, making it unclear how to use this algorithm for other problems with structure.

The purpose of this work is to both unify and improve upon prior ideas for exploiting structure in convex optimization. We begin by formalizing the concept of "structure" using a novel problem

formulation: the minimization of a sum of many piecewise functions. Each piecewise function is defined by multiple simpler subfunctions, at least one of which we assume to be linear. With this formulation, exploiting structure amounts to selectively replacing piecewise terms in the objective with corresponding linear subfunctions. The resulting objective can be considerably simpler to solve.

Using our piecewise formulation, we first present a general theoretical result on exploiting structure in optimization. This result guarantees quantifiable progress toward a problem's global solution by minimizing a simplified objective. We apply this result to derive a new working set algorithm that compares favorably to [11] in that (i) our algorithm results from a minimax optimization of new bounds, and (ii) our algorithm is not limited to constrained objectives. Later, we derive a state-of-the-art screening test by applying the same initial theoretical result. Compared to prior screening tests, our screening result is more effective at simplifying the objective function. Moreover, unlike previous screening results, our screening test applies to a broad class of objectives.

We include empirical evaluations that compare the scalability of screening and working set methods on real-world problems. While many screening tests have been proposed for large-scale optimization, we have not seen the scalability of screening studied in prior literature. Surprisingly, although our screening test significantly improves upon many prior results, we find that screening scales poorly as the size of the problem increases. In fact, in many cases, screening has *negligible* effect on overall convergence times. In contrast, our working set algorithm improves convergence times considerably in a number of cases. This result suggests that compared to screening, working set algorithms are significantly more useful for scaling optimization to large problems.

## 2   Piecewise linear optimization framework

We consider optimization problems of the form
$$\underset{\mathbf{x}\in\mathbb{R}^n}{\text{minimize}}\ f(\mathbf{x}) := \psi(\mathbf{x}) + \sum_{i=1}^{m} \phi_i(\mathbf{x})\,, \tag{P}$$
where $\psi$ is $\gamma$-strongly convex, and each $\phi_i$ is convex and *piecewise*; for each $\phi_i$, we assume a function $\pi_i : \mathbb{R}^n \to \{1, 2, \ldots, p_i\}$ and convex subfunctions $\phi_i^1, \ldots, \phi_i^{p_i}$ such that $\forall \mathbf{x} \in \mathbb{R}^n$, we have
$$\phi_i(\mathbf{x}) = \phi_i^{\pi_i(\mathbf{x})}(\mathbf{x})\,.$$
As will later become clear, we focus on instances of (P) for which many of the subfunctions $\phi_i^k$ are linear. We denote by $\mathcal{X}_i^k$ the subset of $\mathbb{R}^n$ corresponding to the $k$th piecewise subdomain of $\phi_i$:
$$\mathcal{X}_i^k := \{\mathbf{x}\ :\ \pi_i(\mathbf{x}) = k\}\,.$$

The purpose of this work is to develop efficient and principled methods for solving (P) by exploiting the piecewise structure of $f$. Our approach is based on the following observation:

**Proposition 2.1** (Exploiting piecewise structure at $\mathbf{x}^\star$)**.** *Let $\mathbf{x}^\star$ be the minimizer of $f$. For each $i \in [m]$, assume knowledge of $\pi_i(\mathbf{x}^\star)$ and whether $\mathbf{x}^\star \in \text{int}(\mathcal{X}_i^{\pi_i(\mathbf{x}^\star)})$. Define*
$$\phi_i^\star = \left\{ \begin{array}{ll} \phi_i^{\pi_i(\mathbf{x}^\star)} & \text{if } \mathbf{x}^\star \in \text{int}(\mathcal{X}_i^{\pi_i(\mathbf{x}^\star)})\,, \\ \phi_i & \text{otherwise}\,, \end{array} \right.$$
*where $\text{int}(\cdot)$ denotes the interior of a set. Then $\mathbf{x}^\star$ is also the solution to*
$$\underset{\mathbf{x}\in\mathbb{R}^n}{\text{minimize}}\ f^\star(\mathbf{x}) := \psi(\mathbf{x}) + \sum_{i=1}^{m} \phi_i^\star(\mathbf{x})\,. \tag{P$^\star$}$$

In words, Proposition 2.1 states that if $\mathbf{x}^\star$ does not lie on the boundary of the subdomain $\mathcal{X}_i^{\pi_i(\mathbf{x}^\star)}$, then replacing $\phi_i$ with the subfunction $\phi_i^{\pi_i(\mathbf{x}^\star)}$ in $f$ does not affect the minimizer of $f$.

Despite having identical solutions, solving (P$^\star$) can require far less computation than solving (P). This is especially true when many $\phi_i^\star$ are linear, since the sum of linear functions is also linear. More formally, consider a set $\mathcal{W}^\star \subseteq [m]$ such that $\forall i \notin \mathcal{W}^\star$, $\phi_i^\star$ is linear, meaning $\phi_i^\star(\mathbf{x}) = \langle \mathbf{a}_i^\star, \mathbf{x} \rangle + b_i^\star$ for some $\mathbf{a}_i^\star$ and $b_i^\star$. Defining $\mathbf{a}^\star = \sum_{i \notin \mathcal{W}^\star} \mathbf{a}_i^\star$ and $b^\star = \sum_{i \notin \mathcal{W}^\star} b_i^\star$, then (P$^\star$) is equivalent to
$$\underset{\mathbf{x}\in\mathbb{R}^n}{\text{minimize}}\ f^\star(\mathbf{x}) := \psi(\mathbf{x}) + \langle \mathbf{a}^\star, \mathbf{x} \rangle + b^\star + \sum_{i \in \mathcal{W}^\star} \phi_i^\star(\mathbf{x})\,. \tag{P$^{\star\star}$}$$

That is, (P) has been reduced from a problem with $m$ piecewise functions to a problem of size $|\mathcal{W}^\star|$. Since often $|\mathcal{W}^\star| \ll m$, solving (P$^\star$) can be tremendously simpler than solving (P). The scenario is quite common in machine learning applications. Some important examples include:

- *Piecewise loss minimization:* $\phi_i$ is a piecewise loss with at least one linear subfunction.
- *Constrained optimization:* $\phi_i$ takes value 0 for a subset of $\mathbb{R}^n$ and $+\infty$ otherwise.
- *Optimization with sparsity inducing penalties:* $\ell_1$-regularized regression, group lasso, fused lasso, etc., are instances of (P) via duality [13].

We include elaboration on these examples in Appendix A.

## 3 Theoretical results

We have seen that solving (P$^\star$) can be more efficient than solving (P). However, since $\mathcal{W}^\star$ is unknown prior to optimization, solving (P$^\star$) is impractical. Instead, we can hope to design algorithms that rapidly learn $\mathcal{W}^\star$. In this section, we propose principled methods for achieving this goal.

### 3.1 A general mechanism for exploiting piecewise linear structure

In this section, we focus on the consequences of minimizing the function
$$f'(\mathbf{x}) := \psi(\mathbf{x}) + \sum_{i=1}^{m} \phi_i'(\mathbf{x}),$$
where $\phi_i' \in \{\phi_i\} \cup \{\phi_i^1, \ldots, \phi_i^{p_i}\}$. That is, $\phi_i'$ is either the original piecewise function $\phi_i$ or one of its subfunctions $\phi_i^k$. With (P$^\star$) unknown, it is natural to consider this more general class of objectives (in the case that $\phi_i' = \phi_i^\star$ for all $i$, we see $f'$ is the objective function of (P$^\star$)). The goal of this section is to establish choices of $f'$ such that by minimizing $f'$, we can make progress toward minimizing $f$. We later introduce working set and screening methods based on this result.

To guide the choice of $f'$, we assume points $\mathbf{x}_0 \in \mathbb{R}^n, \mathbf{y}_0 \in \mathrm{dom}(f)$, where $\mathbf{x}_0$ minimizes a $\gamma$-strongly convex function $f_0$ that lower bounds $f$. The point $\mathbf{y}_0$ represents an existing approximation of $\mathbf{x}^\star$, while $\mathbf{x}_0$ can be viewed as a second approximation related to a point in (P)'s dual space. Since $f_0$ lower bounds $f$ and $\mathbf{x}_0$ minimizes $f_0$, note that $f_0(\mathbf{x}_0) \leq f_0(\mathbf{x}^\star) \leq f(\mathbf{x}^\star)$. Using this fact, we quantify the suboptimality of $\mathbf{x}_0$ and $\mathbf{y}_0$ in terms of the suboptimality gap
$$\Delta_0 := f(\mathbf{y}_0) - f_0(\mathbf{x}_0) \geq f(\mathbf{y}_0) - f(\mathbf{x}^\star). \tag{1}$$
Importantly, we consider choices of $f'$ such that by minimizing $f'$, we can form points $(\mathbf{x}', \mathbf{y}')$ that improve upon the existing approximations $(\mathbf{x}_0, \mathbf{y}_0)$ in terms of the suboptimality gap. Specifically, we define $\mathbf{x}'$ as the minimizer of $f'$, while $\mathbf{y}'$ is a point on the segment $[\mathbf{y}_0, \mathbf{x}']$ (to be defined precisely later). Our result in this section applies to choices of $f'$ that satisfy three natural requirements:

R1. *Tight in a neighborhood of $\mathbf{y}_0$:* For a closed set $\mathcal{S}$ with $\mathbf{y}_0 \in \mathrm{int}(\mathcal{S})$, $f'(\mathbf{x}) = f(\mathbf{x}) \ \forall \mathbf{x} \in \mathcal{S}$.
R2. *Lower bound on $f$:* For all $\mathbf{x}$, we have $f'(\mathbf{x}) \leq f(\mathbf{x})$.
R3. *Upper bound on $f_0$:* For all $\mathbf{x}$, we have $f'(\mathbf{x}) \geq f_0(\mathbf{x})$.

Each of these requirements serves a specific purpose. After solving $\mathbf{x}' := \mathrm{argmin}_{\mathbf{x}} f'(\mathbf{x})$, R1 enables a backtracking operation to obtain a point $\mathbf{y}'$ such that $f(\mathbf{y}') < f(\mathbf{y}_0)$ (assuming $\mathbf{y}_0 \neq \mathbf{x}^\star$). We define $\mathbf{y}'$ as the point on the segment $(\mathbf{y}_0, \mathbf{x}']$ that is closest to $\mathbf{x}'$ while remaining in the set $\mathcal{S}$:
$$\theta' := \max \{\theta \in (0, 1] : \theta \mathbf{x}' + (1 - \theta)\mathbf{y}_0 \in \mathcal{S}\}, \quad \mathbf{y}' := \theta'\mathbf{x}' + (1 - \theta')\mathbf{y}_0. \tag{2}$$
Since (i) $f'$ is convex, (ii) $\mathbf{x}'$ minimizes $f'$, and (iii) $\mathbf{y}_0 \in \mathrm{int}(\mathcal{S})$, it follows that $f(\mathbf{y}') \leq f(\mathbf{y}_0)$. Applying R2 leads to the new suboptimality gap
$$\Delta' := f(\mathbf{y}') - f'(\mathbf{x}') \geq f(\mathbf{y}') - f(\mathbf{x}^\star). \tag{3}$$
R2 is also a natural requirement since we are interested in the scenario that many $\phi_i'$ are linear, in which case (i) $\phi_i'$ lower bounds $\phi_i$ as a result of convexity, and (ii) the resulting $f'$ likely can be minimized efficiently. Finally, R3 is useful for ensuring $f'(\mathbf{x}') \geq f_0(\mathbf{x}') \geq f_0(\mathbf{x}_0)$. It follows that $\Delta' \leq \Delta_0$. Moreover, this improvement in suboptimality gap can be quantified as follows:

**Lemma 3.1** (Guaranteed suboptimality gap progress—proven in Appendix B). *Consider points* $\mathbf{x}_0 \in \mathbb{R}^n, \mathbf{y}_0 \in \mathrm{dom}(f)$ *such that* $\mathbf{x}_0$ *minimizes a* $\gamma$-*strongly convex function* $f_0$ *that lower bounds* $f$. *For any function* $f'$ *that satisfies R1, R2, and R3, let* $\mathbf{x}'$ *be the minimizer of* $f'$, *and define* $\theta'$ *and* $\mathbf{y}'$ *via backtracking as in (2). Then defining suboptimality gaps* $\Delta_0$ *and* $\Delta'$ *as in (1) and (3), we have*

$$\Delta' \leq (1 - \theta') \left[ \Delta_0 - \tfrac{1+\theta'}{\theta'^2} \tfrac{\gamma}{2} \min_{\mathbf{z} \notin \mathrm{int}(\mathcal{S})} \left\| \mathbf{z} - \tfrac{\theta'\mathbf{x}_0 + \mathbf{y}_0}{1+\theta'} \right\|^2 - \tfrac{\theta'}{1+\theta'} \tfrac{\gamma}{2} \|\mathbf{x}_0 - \mathbf{y}_0\|^2 \right].$$

The primary significance of Lemma 3.1 is the bound's relatively simple dependence on $\mathcal{S}$. We next design working set and screening methods that choose $\mathcal{S}$ to optimize this bound.

---

**Algorithm 1** PW-BLITZ

---
**initialize** $\mathbf{y}_0 \in \mathrm{dom}(f)$
*# Initialize $\mathbf{x}_0$ by minimizing a simple lower bound on $f$:*
$\forall i \in [m], \phi'_{i,0}(\mathbf{x}) := \phi_i(\mathbf{y}_0) + \langle \mathbf{g}_i, \mathbf{x} - \mathbf{y}_0 \rangle$, where $\mathbf{g}_i \in \partial \phi_i(\mathbf{y}_0)$
$\mathbf{x}_0 \leftarrow \mathrm{argmin}_{\mathbf{x}} f'_0(\mathbf{x}) := \psi(\mathbf{x}) + \sum_{i=1}^{m} \phi'_{i,0}(\mathbf{x})$
**for** $t = 1, \ldots, T$ **until** $\mathbf{x}_T = \mathbf{y}_T$ **do**
    *# Form subproblem*:
    Select $\beta_t \in [0, \frac{1}{2}]$
    $\mathbf{c}_t \leftarrow \beta_t \mathbf{x}_{t-1} + (1 - \beta_t) \mathbf{y}_{t-1}$
    Select threshold $\tau_t > \beta_t \| \mathbf{x}_{t-1} - \mathbf{y}_{t-1} \|$
    $\mathcal{S}_t := \{ \mathbf{x} \, : \, \| \mathbf{x} - \mathbf{c}_t \| \leq \tau_t \}$
    **for** $i = 1, \ldots, m$ **do**
        $k \leftarrow \pi_i(\mathbf{y}_{t-1})$
        **if** (C1 **and** C2 **and** C3) **then** $\phi'_{i,t} := \phi_i^k$ **else** $\phi'_{i,t} := \phi_i$
    *# Solve subproblem*:
    $\mathbf{x}_t \leftarrow \mathrm{argmin}_{\mathbf{x}} f'_t(\mathbf{x}) := \psi(\mathbf{x}) + \sum_{i=1}^{m} \phi'_{i,t}(\mathbf{x})$
    *# Backtrack*:
    $\alpha_t \leftarrow \mathrm{argmin}_{\alpha \in (0,1]} f(\alpha \mathbf{x}_t + (1 - \alpha) \mathbf{y}_{t-1})$
    $\mathbf{y}_t \leftarrow \alpha_t \mathbf{x}_t + (1 - \alpha_t) \mathbf{y}_{t-1}$
**return** $\mathbf{y}_T$

---

### 3.2 Piecewise working set algorithm

Lemma 3.1 suggests an iterative algorithm that, at each iteration $t$, minimizes a modified objective $f'_t(\mathbf{x}) := \psi(\mathbf{x}) + \sum_{i=1}^{m} \phi'_{i,t}(\mathbf{x})$, where $\phi'_{i,t} \in \{\phi_i\} \cup \{\phi_i^1, \ldots, \phi_i^{p_i}\}$. To guide the choice of each $\phi'_{i,t}$, our algorithm considers previous iterates $\mathbf{x}_{t-1}$ and $\mathbf{y}_{t-1}$, where $\mathbf{x}_{t-1}$ minimizes $f'_{t-1}$. For all $i \in [m], j = \phi_i(\mathbf{y}_{t-1})$, we define $\phi'_{i,t} = \phi_i^j$ if the following three conditions are satisfied:

C1. *Tight in the neighborhood of $\mathbf{y}_{t-1}$*: We have $\mathcal{S}_t \subseteq \mathcal{X}_i^k$ (implying $\phi_i(\mathbf{x}) = \phi_i^k(\mathbf{x}) \, \forall \mathbf{x} \in \mathcal{S}_t$).
C2. *Lower bound on $\phi_i$*: For all $\mathbf{x}$, we have $\phi_i^k(\mathbf{x}) \leq \phi_i(\mathbf{x})$.
C3. *Upper bound on $\phi'_{i,t-1}$ in the neighborhood of $\mathbf{x}_{t-1}$*: For all $\mathbf{x} \in \mathbb{R}^n$ and $\mathbf{g}_i \in \partial \phi'_{i,t-1}(\mathbf{x}_{t-1})$, we have $\phi_i^k(\mathbf{x}) \geq \phi'_{i,t-1}(\mathbf{x}_{t-1}) + \langle \mathbf{g}_i, \mathbf{x} - \mathbf{x}_{t-1} \rangle$.

If any of the above conditions are unmet, then we let $\phi'_{i,t} = \phi_i$. As detailed in Appendix C, this choice of $\phi'_{i,t}$ ensures $f'_t$ satisfies conditions analogous to conditions R1, R2, and R3 for Lemma 3.1.

After determining $f'_t$, the algorithm proceeds by solving $\mathbf{x}_t \leftarrow \mathrm{argmin}_{\mathbf{x}} f'_t(\mathbf{x})$. We then set $\mathbf{y}_t \leftarrow \alpha_t \mathbf{x}_t + (1 - \alpha_t) \mathbf{y}_{t-1}$, where $\alpha_t$ is chosen via backtracking. Lemma 3.1 implies the sub-optimality gap $\Delta_t := f(\mathbf{y}_t) - f'_t(\mathbf{x}_t)$ decreases with $t$ until $\mathbf{x}_T = \mathbf{y}_T$, at which point $\Delta_T = 0$ and $\mathbf{x}_T$ and $\mathbf{y}_T$ solve (P). Defined in Algorithm 1, we call this algorithm "PW-BLITZ" as it extends the BLITZ algorithm for constrained problems from [11] to a broader class of piecewise objectives.

An important consideration of Algorithm 1 is the choice of $\mathcal{S}_t$. If $\mathcal{S}_t$ is large, C1 is easily violated, meaning $\phi'_{i,t} = \phi_i$ for many $i$. This implies $f'_t$ is difficult to minimize. In contrast, if $\mathcal{S}_t$ is small, then $\phi'_{i,t}$ is potentially linear for many $i$. In this case, $f'_t$ is simpler to minimize, but $\Delta_t$ may be large.

Interestingly, conditioned on oracle knowledge of $\theta_t := \max \{\theta \in (0, 1] \, : \, \theta \mathbf{x}_t + (1 - \theta) \mathbf{y}_{t-1} \in \mathcal{S}_t\}$, we can derive an optimal $\mathcal{S}_t$ according to Lemma 3.1 subject to a volume constraint $\mathrm{vol}(\mathcal{S}_t) \leq V$:

$$\mathcal{S}_t^\star := \underset{\mathcal{S} \, : \, \mathrm{vol}(\mathcal{S}) \leq V}{\mathrm{argmax}} \, \min_{\mathbf{z} \notin \mathrm{int}(\mathcal{S})} \left\| \mathbf{z} - \tfrac{\theta_t \mathbf{x}_{t-1} + \mathbf{y}_{t-1}}{1 + \theta_t} \right\|.$$

$\mathcal{S}_t^\star$ is a ball with center $\frac{\theta_t \mathbf{x}_{t-1} + \mathbf{y}_{t-1}}{1 + \theta_t}$. Of course, this result cannot be used in practice directly, since $\theta_t$ is unknown when choosing $\mathcal{S}_t$. Motivated by this result, Algorithm 1 instead defines $\mathcal{S}_t$ as a ball with radius $\tau_t$ and a similar center $\mathbf{c}_t := \beta_t \mathbf{x}_{t-1} + (1 - \beta_t) \mathbf{y}_{t-1}$ for some $\beta_t \in [0, \frac{1}{2}]$.

By choosing $\mathcal{S}_t$ in this manner, we can quantify the amount of progress Algorithm 1 makes at ieration $t$. Our first theorem lower bounds the amount of progress during iteration $t$ of Algorithm 1 for the case in which $\beta_t$ happens to be chosen optimally. That is, $\mathcal{S}_t$ is a ball with center $\frac{\theta_t \mathbf{x}_{t-1} + \mathbf{y}_{t-1}}{1+\theta_t}$.

**Theorem 3.2** (Convergence progress with optimal $\beta_t$)**.** *Let $\Delta_{t-1}$ and $\Delta_t$ be the suboptimality gaps after iterations $t-1$ and $t$ of Algorithm 1, and suppose that $\beta_t = \theta_t(1+\theta_t)^{-1}$. Then*

$$\Delta_t \leq \Delta_{t-1} + \tfrac{\gamma}{2}\tau_t^2 - \tfrac{3}{2}\left(\gamma\tau_t^2\Delta_{t-1}^2\right)^{1/3}.$$

Since the optimal $\beta_t$ is unknown when choosing $\mathcal{S}_t$, our second theorem characterizes the worst-case performance of extremal choices of $\beta_t$ (the cases $\beta_t = 0$ and $\beta_t = \frac{1}{2}$).

**Theorem 3.3** (Convergence progress with suboptimal $\beta_t$)**.** *Let $\Delta_{t-1}$ and $\Delta_t$ be the suboptimality gaps after iterations $t-1$ and $t$ of Algorithm 1, and suppose that $\beta_t = 0$. Then*

$$\Delta_t \leq \Delta_{t-1} + \tfrac{\gamma}{2}\tau_t^2 - (2\gamma\tau_t^2\Delta_{t-1})^{1/2}.$$

*Alternatively, suppose that $\beta_t = \frac{1}{2}$ , and define $d_t := \|\mathbf{x}_{t-1} - \mathbf{y}_{t-1}\|$. Then*

$$\Delta_t \leq \Delta_{t-1} + \tfrac{\gamma}{2}(\tau_t - \tfrac{1}{2}d_t)^2 - \tfrac{3}{2}\left(\gamma(\tau_t - \tfrac{1}{2}d_t)^2\Delta_{t-1}^2\right)^{1/3}.$$

These results are proven in Appendices D and E. Note that it is often desirable to choose $\tau_t$ such that $\frac{\gamma}{2}\tau_t^2$ is significantly less than $\Delta_{t-1}$. (In the alternative case, the subproblem objective $f_t'$ may be no simpler than $f$. One could choose $\tau_t$ such that $\Delta_t = 0$, for example, but as we will see in §3.3, we are only performing screening in this scenario.) Assuming $\frac{\gamma}{2}\tau_t^2$ is small in relation to $\Delta_{t-1}$, the ability to choose $\beta_t$ is advantageous in terms of worst case bounds if one manages to select $\beta_t \approx \theta_t(1+\theta_t)^{-1}$. At the same time, Theorem 3.3 suggests that Algorithm 1 is robust to the choice of $\beta_t$; the algorithm makes progress toward convergence even with worst-case choices of this parameter.

**Practical considerations**   We make several notes about using Algorithm 1 in practice. Since subproblem solvers are iterative, it is important to only compute $\mathbf{x}_t$ approximately. In Appendix F, we include a modified version of Lemma 3.1 that considers this case. This result suggests terminating subproblem $t$ when $f_t'(\mathbf{x}_t) - \min_{\mathbf{x}} f_t'(\mathbf{x}) \leq \epsilon\Delta_{t-1}$ for some $\epsilon \in (0,1)$. Here $\epsilon$ trades off the amount of progress resulting from solving subproblem $t$ with the time dedicated to solving this subproblem.

To choose $\beta_t$, we find it practical to initialize $\beta_0 = 0$ and let $\beta_t = \alpha_{t-1}(1+\alpha_{t-1})^{-1}$ for $t > 0$. This roughly approximates the optimal choice $\beta_t = \theta_t(1+\theta_t)^{-1}$, since $\theta_t$ can be viewed as a worst-case version of $\alpha_t$, and $\alpha_t$ often changes gradually with $t$. Selecting $\tau_t$ is problem dependent. By letting $\tau_t = \beta_t \|\mathbf{x}_{t-1} - \mathbf{y}_{t-1}\| + \xi\Delta_{t-1}^{1/2}$ for a small $\xi > 0$, Algorithm 1 converges linearly in $t$. It can also be beneficial to choose $\tau_t$ in other ways—for example, choosing $\tau_t$ so subproblem $t$ fits in memory.

It is also important to recognize the relative amount of time required for each stage of Algorithm 1. When forming subproblem $t$, the time consuming step is checking condition C1. In the most common scenarios that $\mathcal{X}_i^k$ is a half-space or ball, this condition is testable in $\mathcal{O}(n)$ time. However, for arbitrary regions, this condition could be difficult to test. The time required for solving subproblem $t$ is clearly application dependent, but we note it can be helpful to select subproblem termination criteria to balance time usage between stages of the algorithm. The backtracking stage is a 1D convex problem that at most requires evaluating $f$ a logarithmic number of times. Simpler backtracking approaches are available for many objectives. It is also not necessary to perform exact backtracking.

**Relation to BLITZ algorithm**   Algorithm 1 is related to the BLITZ algorithm [11]. BLITZ applies only to constrained problems, however, while Algorithm 1 applies to a more general class of piecewise objectives. In Appendix G, we ellaborate on Algorithm 1's connection to BLITZ and other algorithms.

## 3.3   Piecewise screening test

Lemma 3.1 can also be used to simplify the objective $f$ in such a way that the minimizer $\mathbf{x}^\star$ is unchanged. Recall Lemma 3.1 assumes a function $f'$ and set $\mathcal{S}$ for which $f'(\mathbf{x}) = f(\mathbf{x})$ for all $\mathbf{x} \in \mathcal{S}$. The idea of this section is to select the smallest region $\mathcal{S}$ such that in Lemma 3.1, $\Delta'$ must equal 0 (according to the lemma). In this case, the minimizer of $f'$ is equal to the minimizer of $f$—even though $f'$ is potentially much simpler to minimize. This results in the following screening test:

**Theorem 3.4** (Piecewise screening test—proven in Appendix H). *Consider any $\mathbf{x}_0, \mathbf{y}_0 \in \mathbb{R}^n$ such that $\mathbf{x}_0$ minimizes a $\gamma$-strongly convex function $f_0$ that lower bounds $f$. Define the suboptimality gap $\Delta_0 := f(\mathbf{y}_0) - f_0(\mathbf{x}_0)$ as well as the point $\mathbf{c}_0 := \frac{\mathbf{x}_0 + \mathbf{y}_0}{2}$. Then for any $i \in [m]$ and $k = \pi_i(\mathbf{y}_0)$, if*

$$\mathcal{S} := \left\{ \mathbf{x} \ : \ \|\mathbf{x} - \mathbf{c}_0\| \leq \sqrt{\tfrac{1}{\gamma}\Delta_0 - \tfrac{1}{4}\|\mathbf{x}_0 - \mathbf{y}_0\|^2} \right\} \subseteq \mathrm{int}(\mathcal{X}_i^k),$$

*then $\mathbf{x}^\star \in \mathrm{int}(\mathcal{X}_i^k)$. This implies $\phi_i$ may be replaced with $\phi_i^k$ in (P) without affecting $\mathbf{x}^\star$.*

Theorem 3.4 applies to general $\mathcal{X}_i^k$, and testing if $\mathcal{S} \subseteq \mathrm{int}(\mathcal{X}_i^k)$ may be difficult. Fortunately, $\mathcal{X}_i^k$ often is (or is a superset of) a simple region that makes applying Theorem 3.4 simple.

**Corollary 3.5** (Piecewise screening test for half-space $\mathcal{X}_i^k$). *Suppose that $\mathcal{X}_i^k \supseteq \{\mathbf{x} \ : \ \langle \mathbf{a}_i, \mathbf{x} \rangle \leq b_i\}$ for some $\mathbf{a}_i \in \mathbb{R}^n, b_i \in \mathbb{R}$. Define $\mathbf{x}_0, \mathbf{y}_0, \Delta_0$, and $\mathbf{c}_0$ as in Theorem 3.4. Then $\mathbf{x}^\star \in \mathrm{int}(\mathcal{X}_i^k)$ if*

$$\frac{b_i - \langle \mathbf{a}_i, \mathbf{c}_0 \rangle}{\|\mathbf{a}_i\|} > \sqrt{\tfrac{1}{\gamma}\Delta_0 - \tfrac{1}{4}\|\mathbf{x}_0 - \mathbf{y}_0\|^2}.$$

**Corollary 3.6** (Piecewise screening test for ball $\mathcal{X}_i^k$). *Suppose that $\mathcal{X}_i^k \supseteq \{\mathbf{x} \ : \ \|\mathbf{x} - \mathbf{a}_i\| \leq b_i\}$ for some $\mathbf{a}_i \in \mathbb{R}^n, b_i \in \mathbb{R}_{>0}$. Define $\mathbf{x}_0, \mathbf{y}_0, \Delta_0$, and $\mathbf{c}_0$ as in Theorem 3.4. Then $\mathbf{x}^\star \in \mathrm{int}(\mathcal{X}_i^k)$ if*

$$b_i - \|\mathbf{a}_i - \mathbf{c}_0\| > \sqrt{\tfrac{1}{\gamma}\Delta_0 - \tfrac{1}{4}\|\mathbf{x}_0 - \mathbf{y}_0\|^2}.$$

Corollary 3.5 applies to piecewise loss minimization (for SVMs, discarding examples that are not marginal support vectors), $\ell_1$-regularized learning (discarding irrelevant features), and optimization with linear constraints (discarding superfluous constraints). Applications of Corollary 3.6 include group lasso and many constrained objectives. In order to obtain the point $\mathbf{x}_0$, it is usually practical to choose $f_0$ as the sum of $\psi$ and a first-order lower bound on $\sum_{i=1}^{m} \phi_i$. In this case, computing $\mathbf{x}_0$ is as simple as finding the conjugate of $\psi$. We illustrate this idea with an SVM example in Appendix I.

Since $\Delta_0$ decreases over the course of an iterative algorithm, Theorem 3.4 is "adaptive," meaning it increases in effectiveness as progress is made toward convergence. In contrast, most screening tests are "nonadaptive." Nonadaptive screening tests depend on knowledge of an exact solution to a related problem, which is disadvantageous, since (i) solving a related problem exactly is generally computationally expensive, and (ii) the screening test can only be applied prior to optimization.

**Relation to existing screening tests**  Theorem 3.4 generalizes and improves upon many existing screening tests. We summarize Theorem 3.4's relation to previous results below. Unlike Theorem 3.4, existing tests typically apply to only one or two objectives. Elaboration is included in Appendix J.

- *Adaptive tests for sparse optimization:* Recently, [6], [7], and [8] considered adaptive screening tests for several sparse optimization problems, including $\ell_1$-regularized learning and group lasso. These tests rely on knowledge of primal and dual points (analogous to $\mathbf{x}_0$ and $\mathbf{y}_0$), but the tests are not as effective (nor as general) as Theorem 3.4.
- *Adaptive tests for constrained optimization:* [11] considered screening with primal-dual pairs for constrained optimization problems. The resulting test is a more general version (applies to more objectives) of [6], [7], and [8]. Thus, Theorem 3.4 improves upon [11] as well.
- *Nonadaptive tests for degree 1 homogeneous loss minimization:* [10] considered screening for $\ell_2$-regularized learning with hinge and $\ell_1$ loss functions. This is a special non-adaptive case of Theorem 3.4, which requires solving the problem with greater regularization prior to screening.
- *Nonadaptive tests for sparse optimization:* Some tests, such as [4] for the lasso, may screen components that Theorem 3.4 does not eliminate. In Appendix J, we show how Theorem 3.4 can be modified to generalize [4], but this change increases the time needed for screening. In practice, we were unable to overcome this drawback to speed up iterative algorithms.

**Relation to working set algorithm**  Theorem 3.4 is closely related to Algorithm 1. In particular, our screening test can be viewed as a working set algorithm that converges in one iteration. In the context of Algorithm 1, this amounts to choosing $\beta_1 = \frac{1}{2}$ and $\tau_1 = \sqrt{\tfrac{1}{\gamma}\Delta_0 - \tfrac{1}{4}\|\mathbf{x}_0 - \mathbf{y}_0\|^2}$.

It is important to understand that it is usually not desirable that a working set algorithm converges in one iteration. Since screening rules do not make errors, these methods simplify the objective by only a modest amount. In many cases, screening may fail to simplify the objective in *any* meaningful way. In the following section, we consider real-world scenarios to demonstrate these points.

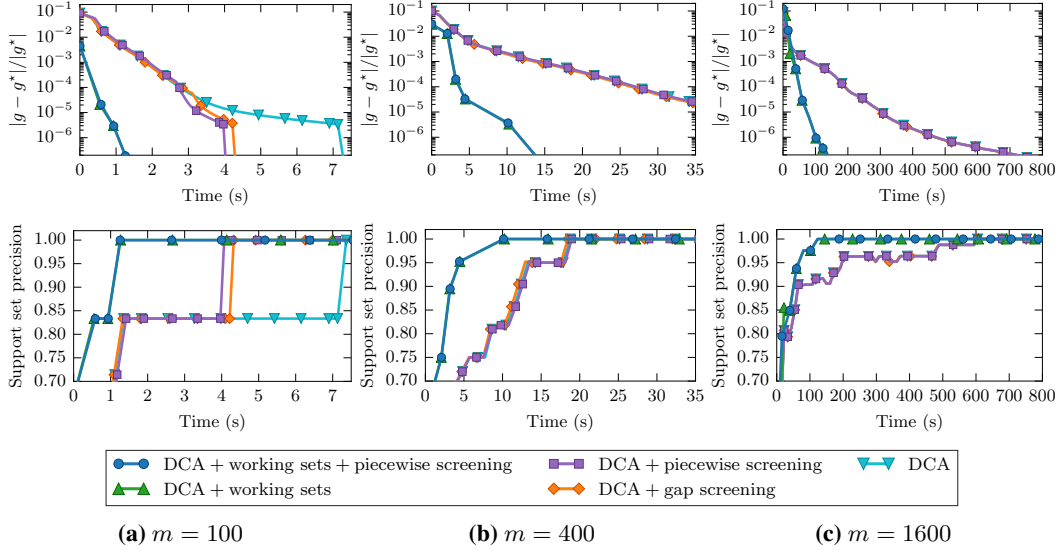

**(a)** $m = 100$        **(b)** $m = 400$        **(c)** $m = 1600$

Figure 1: **Group lasso convergence comparison.** While screening is marginally useful for the problem with only 100 groups, screening becomes ineffective as $m$ increases. The working set algorithm convincingly outperforms dual coordinate descent in all cases.

## 4 Comparing the scalability of screening and working set methods

This section compares the scalability of our working set and screening approaches. We consider two popular instances of (P): group lasso and linear SVMs. For each problem, we examine the performance of our working set algorithm and screening rule as $m$ increases. This is an important comparison, as we have not seen such scalability experiments in prior works on screening.

We implemented dual coordinate ascent (DCA) to solve each instance of (P). DCA is known to be simple and fast, and there are no parameters to tune. We compare DCA to three alternatives:

1. *DCA + screening:* After every five DCA epochs we apply screening. "Piecewise screening" refers to Theorem 3.4. For group lasso, we also implement "gap screening" [7].
2. *DCA + working sets:* Implementation of Algorithm 1. DCA is used to solve each subproblem.
3. *DCA + working sets + screening:* Algorithm 1 with Theorem 3.4 applied after each iteration.

**Group lasso comparisons**    We define the group lasso objective as

$$g_{\text{GL}}(\boldsymbol{\omega}) := \tfrac{1}{2} \left\| \mathbf{A}\boldsymbol{\omega} - \mathbf{b} \right\|^2 + \lambda \sum_{i=1}^{m} \left\| \boldsymbol{\omega}_{\mathcal{G}_i} \right\|_2 .$$

$\mathbf{A} \in \mathbb{R}^{n \times q}$ is a design matrix, and $\mathbf{b} \in \mathbb{R}^n$ is a labels vector. $\lambda > 0$ is a regularization parameter, and $\mathcal{G}_1, \dots, \mathcal{G}_m$ are disjoint sets of feature indices such that $\cup_{i=1}^m \mathcal{G}_i = [q]$. Denote a minimizer of $g_{\text{GL}}$ by $\boldsymbol{\omega}^\star$. For large $\lambda$, groups of elements, $\boldsymbol{\omega}_{\mathcal{G}_i}^\star$, have value 0 for many $\mathcal{G}_i$. While $g_{\text{GL}}$ is not directly an instance of (P), the dual of $g_{\text{GL}}$ is strongly concave with $m$ constraints (and thus an instance of (P)).

We consider an instance of $g_{\text{GL}}$ to perform feature selection for an insurance claim prediction task[1]. Given $n = 250{,}000$ training instances, we learned an ensemble of 1600 decision trees. To make predictions more efficiently, we use group lasso to reduce the number of trees in the model. The resulting problem has $m = 1600$ groups and $q = 28{,}733$ features. To evaluate the dependence of the algorithms on $m$, we form smaller problems by uniformly subsampling 100 and 400 groups. For each problem we set $\lambda$ so that exactly 5% of groups have nonzero weight in the optimal model.

Figure 1 contains results of this experiment. Our metrics include the relative suboptimality of the current iterate as well as the agreement of this iterate's nonzero groups with those of the optimal solution in terms of precision (all algorithms had high recall). This second metric is arguably more important, since the task is feature selection. Our results illustrate that while screening is marginally helpful when $m$ is small, our working set method is more effective when scaling to large problems.

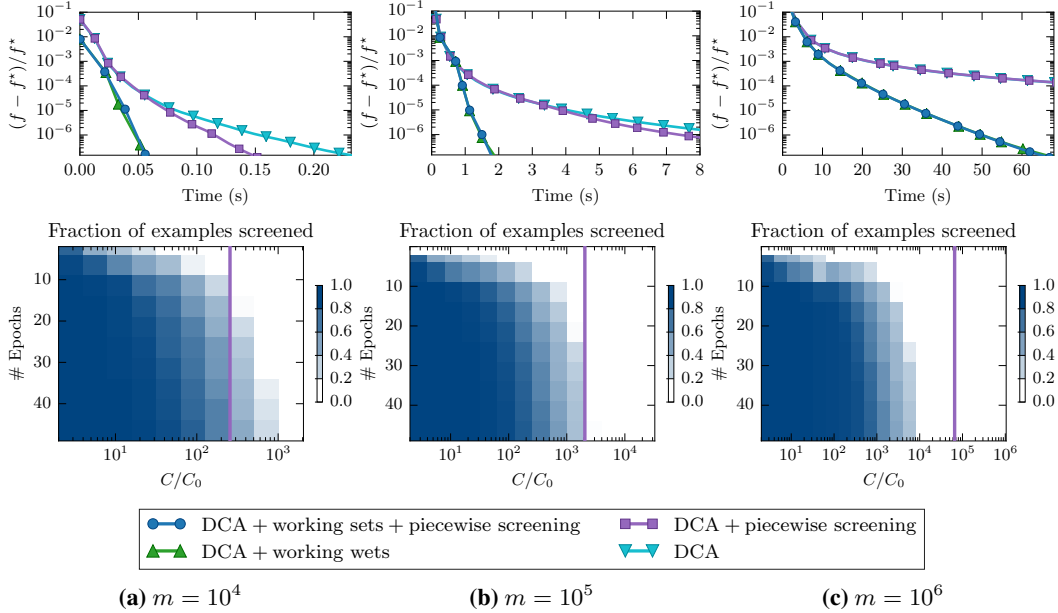

**(a)** $m = 10^4$  **(b)** $m = 10^5$  **(c)** $m = 10^6$

Figure 2: **SVM convergence comparison. (above)** Relative suboptimality vs. time. **(below)** Heat map depicting fraction of examples screened by Theorem 3.4 when used in conjunction with dual coordinate ascent. $y$-axis is the number of epochs completed; $x$-axis is the tuning parameter $C$. $C_0$ is the largest value of $C$ for which each element of the dual solution takes value $C$. Darker regions indicate more successful screening. The vertical line indicates the choice of $C$ that minimizes validation loss—this is also the choice of $C$ for the above plots. As the number of examples increases, screening becomes progressively less effective near the desirable choice of $C$.

**SVM comparisons**    We define the linear SVM objective as

$$f_{\text{SVM}}(\mathbf{x}) := \tfrac{1}{2} \|\mathbf{x}\|^2 + C \sum_{i=1}^m (1 - b_i \langle \mathbf{a}_i, \mathbf{x} \rangle)_+ .$$

Here $C$ is a tuning parameter, while $\mathbf{a}_i \in \mathbb{R}^n$, $b_i \in \{-1, +1\}$ represents the $i$th training instance. We train an SVM model on the Higgs boson dataset[2]. This dataset was generated by a team of particle physicists. The classification task is to determine whether an event corresponds to the Higgs boson. In order to learn an accurate model, we performed feature engineering on this dataset, resulting in 8010 features. In this experiment, we consider subsets of examples with size $m = 10^4$, $10^5$, and $10^6$.

Results of this experiment are shown in Figure 2. For this problem, we plot the relative suboptimality in terms of objective value. We also include a heat map that shows screening's effectiveness for different values of $C$. Similar to the group lasso results, the utility of screening decreases as $m$ increases. Meanwhile, working sets significantly improve convergence times, regardless of $m$.

## 5   Discussion

Starting from a broadly applicable problem formulation, we have derived principled and unified methods for exploiting piecewise structure in convex optimization. In particular, we have introduced a versatile working set algorithm along with a theoretical understanding of the progress this algorithm makes with each iteration. Using the same analysis, we have also proposed a screening rule that improves upon many prior screening results as well as enables screening for many new objectives.

Our empirical results highlight a significant disadvantage of using screening: unless a good approximate solution is already known, screening is often ineffective. This is perhaps understandable, since screening rules operate under the constraint that erroneous simplifications are forbidden. Working set algorithms are not subject to this constraint. Instead, working set algorithms achieve fast convergence times by aggressively simplifying the objective function, correcting for mistakes only as needed.

**Acknowledgments**

We thank Hyunsu Cho, Christopher Aicher, and Tianqi Chen for their helpful feedback as well as assistance preparing datasets used in our experiments. This work is supported in part by PECASE N00014-13-1-0023, NSF IIS-1258741, and the TerraSwarm Research Center 00008169.

## Footnotes

[1] https://www.kaggle.com/c/ClaimPredictionChallenge

[2]https://archive.ics.uci.edu/ml/datasets/HIGGS

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
