[Reviews · NeurIPS 2016]

Reviewer 1

Summary

The authors propose new algorithms for minimizing composite objectives comprised of strongly convex and piecewise (?) components. Apart from numerous claims about importance of the paper contribution, the authors propose a “working set” algorithm and a piecewise screening test, along with an experimental study which seems to show good performance in some application scenarios.

Qualitative Assessment

The authors seem to be unaware of the literature about bundle methods (see [Hiriart-Urruty, Lemaréchal, 1993] for the references). The proposed “working set” algorithm looks like a bundle family method. Note that the “basic” bundle algorithm – the Kelley method – also often shows fast convergence. Nevertheless, it minimax complexity is disastrous (although in the strongly convex situation it exhibits dimension-independent convergence rates, which are suboptimal). Same as a majority of work on bundle algorithms, the authors provide some bounds for the model gap, but do not seem being interested in providing any guaranties. My feeling is that to legitimate this type of methods (which “often” exhibit “fast convergence”), the minimax analysis is necessary (cf. methods of bundle-level family, [Lemarechal et al, 1995, Lan 2015]. Hiriart-Urruty, J. B., & Lemaréchal, C. (1993). Convex analysis and minimization algorithms II: Advanced theory and bundle methods, vol. 306 of Grundlehren der mathematischen Wissenschaften. Lemaréchal, C., Nemirovskii, A., & Nesterov, Y. (1995). New variants of bundle methods. Mathematical programming, 69(1-3), 111-147. Lan, G. (2015). Bundle-level type methods uniformly optimal for smooth and nonsmooth convex optimization. Mathematical Programming, 149(1-2), 1-45.

Confidence in this Review

1-Less confident (might not have understood significant parts)


Reviewer 2

Summary

The paper studied the problem of selecting important components to solve some particular convex optimization problems faster. Two selection methods are considered to exploit the structure. Two methods for exploiting structures are considered. One is screening methods, the other is working set algorithms. The theoretical analysis of the two methods are unified to solving the particular problem, i.e., minimization of a sum of piecewise functions. The algorithms based on working set technique and screening test are proposed and experimentally tested.

Qualitative Assessment

Overall, this paper is well rewritten. The problems and contributions have been clearly identified. The theoretical and experimental results have been clearly presented. 1. The paper proposed the piecewise optimization framework which enable the unified theoretical analysis of working set and screening tests possible. 2. Theoretical results have been established for the proposed working set algorithms.

Confidence in this Review

2-Confident (read it all; understood it all reasonably well)


Reviewer 3

Summary

This papers proposes a working set algorithm for solving a sum of piecewise functions. Based on it, the author introduces a piecewise (in optimization) screening test.

Qualitative Assessment

Algorithm 1 is a generalization of BLITZ (working set algorithm for sparse optimization). One advance is that it can handle more formulations e.g. the objective of SVM ( l2 regularization hinge loss ). But I have a question about the per-iter backtracking line search for selecting y. For certain problems, this step may have closed form solution, but in general the overhead could be large. Besides, as the author mentioned, testing the conditions C1 - C3 can be time consuming (I think C2 & C3 may even be feasible only when the problem has good structure. Please clarity this point.). Selecting the radius tau in every iteration may be challenging, too. This might limit the set of problem Algorithm 1 can apply to. From the experiments, we do see the working set idea improves the results, as the author acknowledged, the effectiveness of piecewise screening is not clear. Algorithms performs roughly the same without or without piecewise screening, especially for larger m. Besides, the group lasso experiment is in a low dimensional setup. It’s more interesting to see compare the performance in high dimensional setting, p >> n.

Confidence in this Review

2-Confident (read it all; understood it all reasonably well)


Reviewer 4

Summary

This paper provides a unified framework to exploit the piecewise structure of a class of optimization problems. Extensive numerical studies are provided to backup the methods.

Qualitative Assessment

This paper provides a unified framework to exploit the piecewise structure of a class of optimization problems. Extensive numerical studies are provided to backup the methods. The concept is somewhat interesting and novel. However, I do think that there are some issues that I hope the authors should address: 1. The presentation is not very user-friendly. The authors assume a "piecewise" structure of $\phi_i$'s. What does piecewise mean here? My understanding is that the authors assume that these functions are linear on the boundary while more complicated in the interior. Can you find a better name to describe it? Could you provide some geometric illustration? This will help authors to understand it. 2. All the theorems in the paper are like Lemmas to me. This is because these theoretical results are more like building blocks towards a main theorem. What I am expecting here is some thing like f(xt) - f^* \le O(1/t). Am I missing something here? Also, the screening results are not intuitive to me. Can the authors present them in a more user-friendly way?

Confidence in this Review

2-Confident (read it all; understood it all reasonably well)


Reviewer 5

Summary

This paper presents an unified framework for solving a class of piecewise convex functions. Two specific algorithms based on working set and screening test are proposed. The proposed methods are validated on group Lasso and linear SVM. This paper is clear written and solid. However, since some structure output prediction problems are special case of the proposed problem formulation, it seems necessary to empirically and theoretically investigate the relationship between the proposed method and existing results of working set methods along those lines, e.g., see ``Cutting-Plane Training of Structural SVMs”.

Qualitative Assessment

In the section of experiments, it would be nice to compare with other related methods which suit for these problems, like cutting plane methods, sub-gradient projection etc. It is not clear why authors only compare the proposed methods using dual coordinate descent on SVM. As author mentioned, the chosen of \beta_{t} is important for practical problems, it would be good to investigate how the performance varies with different updating scheme and values.

Confidence in this Review

2-Confident (read it all; understood it all reasonably well)


Reviewer 6

Summary

This paper presents a meta optimization algorithm that exploits piecewise structures in the cost function. For large scale problems, handling piecewise linear functions can be costly, which can be significantly reduced by identifying the structure. The paper provides a bound on the relative decrease in the gap between the lower bound and the current estimate, which can be useful for analysis. Based on this bound, the authors propose two ideas for the proposed algorithm: piecewise working set and piecewise screening. The paper provide empirical results on two examples; group lasso and SVM, and shows that the piecewise working set algorithm can achieve faster convergence.

Qualitative Assessment

Overall, I think the issue that the paper is tackling is quite interesting. Most important result in the paper is Lemma 3.1, which can be a good guideline for designing a similar algorithm. Parts of the approach seems unclear, though. Since it is a meta algorithm, the choice of beta and tau will depend on the problem, which has to be tuned by user or designed separately for each problem. There is a doubt regarding Algorithm 1. In line 120, C3 is about the "upper bound on \phi' near x_{t-1}," but the inequality is on the entire space, i.e., x \in R^n. Is this possible? If \phi_i^j and \phi'_{i,t-1}=\phi_i^k are the linear functions about different pieces (j \not k), this seems impossible. I'd like to see an explanation in the rebuttal. It seems that the proposed algorithm shares many similar ideas with active set or trust-region approaches. An explanation about the relationship between them would be nice.

Confidence in this Review

1-Less confident (might not have understood significant parts)